# Bacteria-Derived Protein Aggregates Contribute to the Disruption of Host Proteostasis

**DOI:** 10.3390/ijms23094807

**Published:** 2022-04-27

**Authors:** Alyssa C. Walker, Rohan Bhargava, Autumn S. Dove, Amanda S. Brust, Ali A. Owji, Daniel M. Czyż

**Affiliations:** Department of Microbiology and Cell Science, University of Florida, Gainesville, FL 32611, USA; awalk1@ufl.edu (A.C.W.); bhargav.rohan@ufl.edu (R.B.); autumn.dove@ufl.edu (A.S.D.); abrust@ufl.edu (A.S.B.); ali.owji@ufl.edu (A.A.O.)

**Keywords:** *Caenorhabditis elegans*, proteostasis, bacteria, neurodegenerative diseases, protein aggregates, protein conformational disease, butyrate, aminoglycoside, *Pseudomonas aeruginosa*

## Abstract

Neurodegenerative protein conformational diseases are characterized by the misfolding and aggregation of metastable proteins encoded within the host genome. The host is also home to thousands of proteins encoded within exogenous genomes harbored by bacteria, fungi, and viruses. Yet, their contributions to host protein-folding homeostasis, or proteostasis, remain elusive. Recent studies, including our previous work, suggest that bacterial products contribute to the toxic aggregation of endogenous host proteins. We refer to these products as bacteria-derived protein aggregates (BDPAs). Furthermore, antibiotics were recently associated with an increased risk for neurodegenerative diseases, including Parkinson’s disease and amyotrophic lateral sclerosis—possibly by virtue of altering the composition of the human gut microbiota. Other studies have shown a negative correlation between disease progression and antibiotic administration, supporting their protective effect against neurodegenerative diseases. These contradicting studies emphasize the complexity of the human gut microbiota, the gut–brain axis, and the effect of antibiotics. Here, we further our understanding of bacteria’s effect on host protein folding using the model *Caenorhabditis elegans*. We employed genetic and chemical methods to demonstrate that the proteotoxic effect of bacteria on host protein folding correlates with the presence of BDPAs. Furthermore, the abundance and proteotoxicity of BDPAs are influenced by gentamicin, an aminoglycoside antibiotic that induces protein misfolding, and by butyrate, a short-chain fatty acid that we previously found to affect host protein aggregation and the associated toxicity. Collectively, these results increase our understanding of host–bacteria interactions in the context of protein conformational diseases.

## 1. Introduction

Neurodegenerative protein conformational diseases (PCDs) are disorders of protein homeostasis (proteostasis) and subsequent protein folding capacity, hallmarked by the accumulation of metastable proteins into insoluble aggregates [1]. PCDs such as Alzheimer’s, Parkinson’s, Huntington’s, or amyotrophic lateral sclerosis (ALS) cause a massive loss of life and impose enormous social and financial burdens on afflicted individuals and their families [2]. Currently, the number of individuals affected by Alzheimer’s disease exceeds five million and its annual economic cost surpassed 300 billion USD in the United States alone, and both numbers are projected to more than double by 2050 [3,4]. Despite the high prevalence and the financial burden associated with PCDs, their etiology remains largely unknown. As such, there are no effective treatments or cures [5]. Correlational evidence has revealed a link between PCDs and bacteria within the human gut microbiota (HGM) [6,7,8], which could explain their sporadic onset and unpredictable severity. It has been challenging to study the mechanism by which bacteria within the HGM might be affecting PCD pathogenesis due to the complexity of the microbiome. It has been postulated that bacteria directly affect host protein folding [9,10,11], perhaps, as our previous work suggests, through mechanisms that likely include oxidative stress and bacteria-derived protein aggregates (BDPAs) [12]. As the exact bacterial contribution remains elusive, the influence of antibiotics on disease pathogenesis further supports the role of microbes in the pathogenesis of PCDs; however, the effect of antimicrobials on disease progression is ambiguous, as numerous studies report contradicting data [13,14,15,16].

The expression of aggregating proteins is known to disrupt host proteostasis, consequently affecting the folding of any metastable proteins present within the host proteome [17]. Increasing evidence suggests that bacterial products contribute to the disruption of host proteostasis [18]. Among such products, protein aggregates can be secreted by bacteria into the gut and find their way to the nervous system, potentially affecting the onset and progression of PCDs—a mechanism that was observed by injecting exogenous α-synuclein (a neurotoxic amyloid in Parkinson’s disease) into rats’ intestines and detecting its transport into the brain via the vagal nerve [19]. Amyloids are highly ordered protein aggregates made by eukaryotes and prokaryotes [20,21]. Amyloidal proteins have been the focus of studies due to their specific cross-ß sheet structure, which allows for their detection by simple assays such as Congo Red staining [22]. Amyloidal protein aggregates have been recognized for their proteotoxicity, with a particular emphasis placed on human-derived amyloids eliciting PCD phenotypes in vitro and in vivo [19,23,24]. In a limited number of bacterial species, the effect of amyloids has also been investigated for the heterologous seeding of host metastable proteins. For example, FapC, a *Pseudomonas aeruginosa* secreted functional amyloid, was shown to be proteotoxic to the host and cross-seed with host proteins associated with PCDs [25]. Additional studies have demonstrated that *Escherichia coli* curli fibers increase α-synuclein aggregation in rodents and *Caenorhabditis elegans* [11,26]. While bacterial amyloid proteins contribute to the disruption of host proteostasis, much less is known about the effect of non-amyloidal protein aggregates. 

In comparison to amyloids, bacterial protein aggregates that are non-amyloidal and thus not compatible with standard amyloid detection assays are much less studied. More recently, a growing body of evidence has shed light on the contributions of non-amyloidal BDPAs to bacterial virulence and persistence [27], but their influence on host proteostasis remains less understood and was first highlighted in our earlier work [12]. We previously used *C. elegans* expressing polyglutamine (polyQ) tracts as sensors for the protein folding environment to investigate the effect of *E. coli* curli amyloids on polyQ aggregation in *C. elegans*. While *E. coli* mutants lacking CsgA (the major structural subunit of curli) and CsgD (the master regulator of the curli operon) were both deficient in curli production as assessed by Congo Red staining, it was interesting to find that only one mutant, ∆*csgD*, decreased polyQ aggregation in *C. elegans,* indicating a curli-independent induction of polyQ aggregation. A more sensitive protein aggregate detection assay revealed that ∆*csgD*, but not ∆*csgA* cells, were also deficient in total BDPA production, revealing a potential contribution of non-amyloidal BDPAs to host proteostasis. 

In the present study, we employed pharmacological and genetic means to demonstrate the effect of BDPAs on host proteostasis as assessed by *C. elegans* polyQ aggregation. We used a protein aggregation detection assay, ProteoStat, to identify a positive correlation between BDPAs and polyQ aggregation in *C. elegans*. Furthermore, butyrate influenced this correlation, likely through its effect on bacteria and the host; a result that explains our previous data demonstrating that at a low concentration, butyrate increases *P. aeruginosa*-mediated polyQ aggregation in *C. elegans* [12]. Moreover, bacteria deficient in RpoS, a general stress transcription factor that protects cells from misfolded proteins [28], accumulated more BDPAs and increased polyQ aggregation upon colonization of the *C. elegans* intestine. In agreement with BDPAs being significant contributors to the disruption of host proteostasis, we found that gentamicin, an aminoglycoside antibiotic that induces protein misfolding secondary to mistranslation, also resulted in increased BDPAs and consequently polyQ aggregation. Together our data indicate that misfolded or aggregation-prone bacterial proteins directly contribute to the disruption of the host protein folding environment. These results demonstrate that bacterial proteomes are not inert to the host proteome, and that disruption of bacterial proteostasis ultimately affects the host and likely the pathogenesis of PCDs. 

## 2. Results

### 2.1. Bacteria-Mediated Induction of Host PolyQ Aggregation Correlates with the Abundance of BDPAs

Using *C. elegans* expressing fluorescently labeled polyQ tracts as sensors for the protein folding environment, we previously identified the effect of bacteria on host proteostasis in a screen of species commonly found in the HGM [12]. Out of over 20 bacterial candidates tested, we found the most robust enhancers and suppressors of polyQ aggregation in *C. elegans* were *P. aeruginosa* and *Prevotella corporis*, respectively. As such, in the present study, we chose these two strains to investigate the influence of BDPAs on host proteostasis. To accomplish this, we assessed the relative abundance of BDPAs in both strains using ProteoStat, a protein aggregate detection assay that has superior specificity to protein aggregates [29], and correlated BDPA levels with the bacteria-mediated induction of polyQ aggregation. In agreement with our previous data, we found that *P. aeruginosa* significantly induced polyQ aggregation in the intestine relative to the control *E. coli* strain, while *P. corporis* significantly suppressed polyQ aggregation upon intestinal colonization (Figure 1A). The ProteoStat assay revealed that the extent of polyQ aggregation enhancement mediated by these bacteria is directly proportional to the abundance of BDPAs that originate from their respective strain (Figure 1B,C). These results suggest that host proteostasis is affected by BDPAs, emphasizing the importance of the bacterial composition within the HGM. 

### 2.2. Low-Concentration Butyrate Enhances P. aeruginosa BDPAs

We previously found that both exogenous and endogenous butyrate suppressed bacteria-mediated polyQ aggregation and the associated toxicity, but at low concentrations in the presence of specific bacteria including *P. aeruginosa*, butyrate enhanced polyQ aggregation [12]. Such an effect was not observed when worms were colonized with the *E. coli* OP50 control strain; therefore, we wanted to investigate BDPAs as a possible mechanism by which *P. aeruginosa* contributes to the disruption of host proteostasis. To accomplish this, we assessed the abundance of *P. aeruginosa* BDPAs in the absence and presence of butyrate. Our results reveal that while butyrate suppressed intestinal polyQ aggregation in the presence of *E. coli* (Figure 2A) with no significant change in BDPAs (Figure 2B), it enhanced *P. aeruginosa*-mediated polyQ aggregation (Figure 2C) and BDPAs (Figure 2D). It is not clear how butyrate may affect host proteostasis by targeting bacteria, but our further work shows that, unlike *E. coli*, *P. aeruginosa* can utilize butyrate as a carbon energy source, suggesting that its metabolism may play a role in the increased abundance of BDPAs and pathogenicity in the context of host proteostasis (Appendix A). These results further support that the bacterial influence on host proteostasis is mediated, at least in part, by BDPAs. 

### 2.3. Deletion of E. coli RpoS Sigma-Factor Increases BDPAs and PolyQ Aggregation upon Colonization of the C. elegans Gut 

Thus far, our results suggest that bacterial proteomes affect host proteostasis. To further investigate the contribution of bacterial proteins to the stability of host proteostasis, we used an *E. coli* strain deficient in RpoS, a sigma factor involved in protecting against protein misfolding [28]. As predicted, we found that a deletion of the *rpoS* gene results in significantly elevated levels of BDPAs compared to wild-type *E. coli* (Figure 3A). We hypothesized that if bacterial proteins affect the host proteome, then *C. elegans* colonized by *E. coli* ∆*rpoS* should exhibit higher polyQ aggregation. To test this hypothesis, we colonized worms with bacteria carrying the *rpoS* deletion and, indeed, found a significantly higher polyQ aggregation compared to worms colonized by the wild-type strain (Figure 3B). To confirm that ProteoStat is a sufficient readout of BDPA levels, we extracted and separated soluble and insoluble protein fractions from bacterial cells by sodium dodecyl sulfate-polyacrylamide gel electrophoresis (SDS-PAGE) and visualized total protein by staining the gel with Coomassie (Figure 3C), which is an established method to assess protein aggregate levels [30]. The quantification of stained proteins confirmed that the *rpoS* knockout had a significantly greater level of insoluble proteins, supporting our ProteoStat data and the overall hypothesis (Figure 3D). Because RpoS is an upstream regulator of the CsgD global transcriptional regulator, which also regulates the expression of the CsgA curli subunit, we wanted to ensure that its deletion abrogates curli production. As expected, RpoS-deficient *E. coli* cells lacked amyloid proteins (Figure 4A), which was consistent with previous reports [31]; yet, their colonization of the *C. elegans* intestine increased polyQ aggregation, most likely due to the presence of aggregating proteins that were detected by the ProteoStat assay (Figure 4B). 

### 2.4. Gentamicin Increases the Abundance of E. coli BDPA and PolyQ Aggregation in the C. elegans Intestine

Aminoglycoside antibiotics, such as gentamicin, bind to the bacterial 30S ribosomal subunit leading to mistranslation, an increased presence of misfolded proteins, and eventually cell death [32]. We used a sub-lethal concentration of gentamicin to determine if we could increase the presence of misfolded proteins and consequently affect *C. elegans* proteostasis upon intestinal colonization. Initially, we assessed the effect of gentamicin on the presence of BDPAs using the ProteoStat assay. We found that a low and non-bactericidal concentration of gentamicin significantly increased the abundance of BDPAs (Figure 5A and Appendix A). Further experiments demonstrated that colonization of the *C. elegans* intestine in the presence of a sub-lethal gentamicin concentration on solid nematode growth medium (NGM) significantly enhanced polyQ aggregation (Figure 5B and Appendix A). These results are intriguing as they suggest that aminoglycosides directly contribute to the disruption of bacterial proteostasis, ultimately affecting the protein folding environment in the host. Collectively, our results further support that BDPAs influence host proteostasis and likely contribute to the onset and progression of PCDs. 

## 3. Discussion

In the present study, we demonstrated that the abundance of bacterial protein aggregates positively correlates with the extent to which these bacteria induce polyQ aggregation in *C. elegans* upon intestinal colonization. Two distinct bacteria, *P. aeruginosa* and *P. corporis*, either induce or suppress, respectively, polyQ aggregation upon intestinal colonization, depending on the amount of protein aggregates they produce. These results are intriguing as they suggest that the composition of bacterial proteomes can directly affect host proteostasis (Figure 6). While it was long suspected that gut bacteria contribute to the pathogenesis of neurodegenerative diseases, our results reveal a potential mechanism indicating that bacterial non-amyloidal protein aggregates are a major culprit. Previous studies show that bacterial infections can affect cognitive function. For example, *P. aeruginosa* infection in a mouse model led to a loss of memory [33], whereas in humans, this bacterium is linked to cognitive decline and mood disorders [34,35,36]. How exactly *P. aeruginosa* affects its host remains unknown; however, microbial protein aggregates provide a feasible explanation by which these species can affect the stability and function of the host proteome, possibly through heterologous cross-seeding or sequestering the function of the host proteostasis machinery. Introducing protein aggregates into the proteome was previously shown to affect the folding of other destabilized proteins [17]. Furthermore, microbial products, particularly amyloids, have been shown to induce aggregation of host proteins [18]. For example, *P. aeruginosa* rhamnolipids, secreted surfactants, were shown to enhance the aggregation of human α-synuclein [37], and its functional amyloid, FapC, was involved in heterologous cross-seeding with amyloid-beta [25]. Combined with our data, these results may explain why *P. aeruginosa* abundance is overrepresented in patients with PCDs [38,39]. Additionally, *P. aeruginosa* can adhere to the intestinal epithelial barrier and secrete virulence factors that damage the intestinal epithelium [40]. Damage to the gut epithelium can lead to intestinal permeability, known as “leaky gut syndrome”, increasing the chances of bacterial displacement [41]. An increased abundance of bacteria and their products have been found in patients with PCDs, including in the brains of Alzheimer’s disease patients [42,43,44]. In addition to contributing its aggregating proteins and compromising gut permeability, *P. aeruginosa* is also known to trigger the production and release of host amyloid proteins that are directly associated with PCDs [45,46]. With an estimated 8% of the population being asymptomatically colonized by *P. aeruginosa*, this bacterium may be one of the primary silent microbial contributors to neurodegenerative diseases [47].

The mechanisms by which bacteria release protein aggregates and by which they reach the intracellular space of the intestine remain unknown; however, there could be two possible routes. First, the bacteria are mechanically and enzymatically ruptured, leading to a release of their intracellular content that is taken up by the intestinal cells via endocytosis. The intestine is known for its high endocytic activity that facilitates the uptake of nutrients and trafficking of the vitellogenin (yolk protein) [48]. The other possibility is that viable bacteria secrete aggregation-prone proteins or aggregates that are then taken up by the intestine. Further studies targeting these pathways will reveal potential mechanisms. While our study concentrates on *C. elegans* as a discovery model, we see potential parallel mechanisms in humans where circulating protein aggregates are associated with ALS [49]. It is possible that BDPAs could contribute to the repertoire of circulating and disease-causing aggregates. 

Recent studies, including our work, indicate that *Prevotella* spp. provide protection against protein misfolding and aggregation, and a lower abundance of these bacteria in the human gut microbiota is associated with an increased prevalence and pathogenicity of PCDs, including PD, AD, and ALS [50,51,52,53,54,55,56]. Although previous reports suggest that *Prevotella* spp. may protect against oxidative stress, the exact mechanism by which they enhance host proteostasis is not known [57]. Our data indicate that *Prevotella* did not lead to increased polyQ aggregation upon *C. elegans* intestinal colonization because of the low abundance of BDPAs, and together with its aforementioned role in oxidative stress, it may provide protection against proteotoxicity; though, further studies are needed to reveal the exact mechanism. 

Butyrate is a short-chain fatty acid that was shown to be beneficial to host health [58]. Moreover, its benefits extend to protection against PCDs in animal models [59,60,61,62] and humans, as a decreased abundance of butyrate and butyrogenic bacteria has been associated with neurodegenerative diseases [63,64,65]. In a previous study, we found that butyrate suppresses bacteria-induced polyQ aggregation across tissues, but the suppression depended on the bacteria that colonized the gut [12]. For example, ≤50 mM butyrate enhanced polyQ aggregation in the presence of *P. aeruginosa*. Given that the physiological concentration of butyrate in the gut is estimated at 10–20 mM and over 8% of the population is asymptomatically colonized by *P. aeruginosa*, these results may emphasize a significant contributor to the pathogenesis of PCDs [47,66]. Our present results (Figure 2) showing that a low concentration of butyrate increases *P. aeruginosa* BDPAs may explain the increase of polyQ aggregation. While it remains elusive as to why a low concentration of butyrate increases *P. aeruginosa*-induced proteotoxicity in the host, our results demonstrate that these bacteria can utilize butyrate as their sole carbon source (Appendix A), which could contribute to the increased production of BDPAs, and as a result, make them more virulent. Nakanishi et al. found that at low concentrations, butyrate enhanced the virulence of enterohaemorrhagic *E. coli* [67]. Recently, it was shown that low concentrations of short-chain fatty acids increase the growth and metabolic activity of a *Mycobacterium avium* [68]. Our results also emphasize the importance of a balance between commensal butyrogenic bacteria and detrimental microbes that potentially contribute to the disruption of host proteostasis; however, more research needs to be done to determine the exact effect of butyrate on *P. aeruginosa* and to demonstrate if this is the case in vivo. 

We demonstrated that an *E. coli* strain deficient in RpoS, a sigma factor involved in the regulation of stress responses and protein folding in bacteria [28,69], exhibits a higher abundance of BDPAs, which likely contributes to the increase in *C. elegans* polyQ aggregation upon intestinal colonization. RpoS levels significantly increase upon heat shock to provide protection against aggregation-prone proteins [70,71]. Therefore, we expect that the absence of RpoS leads to an increased abundance of destabilized and aggregating proteins, which would explain why we see a significantly higher abundance of *E. coli* BDPAs in the *rpoS* deletion strain (Figure 3 and Figure 4B). In agreement with previously published results [31], our Congo Red staining confirmed that the RpoS deletion strain is deficient in curli, indicating that BDPAs that contribute to the disruption of host proteostasis, as measured by polyQ aggregation, are not specific to the amyloid family. The lack of influence of bacterial amyloids on polyQ aggregation is surprising but consistent with our previous results. However, because bacterial amyloids are known to affect the stability of host aggregation-prone and disease-associated proteins, including α-synuclein and amyloid-beta, the model that we present (Figure 6) is inclusive of all bacterial-derived aggregating proteins, including both amyloid and non-amyloid BDPAs. The contribution of such non-specific bacterial protein aggregates to the stability of host proteostasis is significant as it suggests that the abundance of certain species, such as *P. aeruginosa*, in the human gut may be detrimental, especially in the presence of a low and physiologically relevant concentration of butyrate. 

We used pharmacological means to disrupt protein folding in bacteria to further support our conclusion that bacteria-derived protein aggregates affect host proteostasis. Gentamicin is a broad-spectrum aminoglycoside antibiotic that induces bacterial death by increasing mistranslation, consequently leading to protein aggregation and cell death [32,72,73]. In agreement with all of the results presented here, we observed an enhancement of BDPAs in gentamicin-treated *E. coli*, and we saw a significant gentamicin-dependent increase in polyQ aggregation upon intestinal colonization. These results are intriguing as aminoglycoside antibiotics are commonly used in a clinical setting and could exacerbate neurodegenerative diseases. In addition, the administration of antibiotics induces dysbiosis between commensal bacteria, targeted non-specifically, and detrimental gram-negative bacteria, which are intrinsically resistant. Two recent studies have linked antibiotic use with an increased risk for neurodegenerative diseases [16,74]. While it is difficult to determine whether antibiotics indirectly affect disease pathogenesis through bacteria, or whether it is the contribution of the infection itself, our data suggest that it is likely both; bacteria alone can induce protein misfolding and aggregation in the host, and antibiotics can potentiate this effect by increasing BDPAs. 

Our findings call for more investigation into the specific characteristics of non-amyloidal BDPAs that affect host proteostasis. The identification of specific aggregating proteins could reveal bacteria that are the most robust contributors to the pathogenesis of PCDs. The elimination of these bacteria from the gut microbiome through personalized therapeutics based on the composition of the patient microbiome could offer a new disease management strategy [75].

## 4. Materials and Methods

### 4.1. Bacterial Strains and Culture

A list of bacterial strains used in this study can be found in Appendix A. Bacteria were cultured under different conditions described under each experiment. 

### 4.2. C. elegans Maintenance and Strains

Nematodes were maintained using previously established protocols [76]. All cultures were kept at 22.5 °C. The *C. elegans* strain used in this study can be found in Appendix A. 

### 4.3. Aggregate Quantification

*P. aeruginosa*, *P. corporis*, and *E. coli* OP50 were grown at 37 °C on Tryptic Soy Agar (TSA) plates supplemented with 5% defibrinated sheep blood. *P. corporis* was kept at 37 °C in an anaerobic chamber with AnaeroPacks. Bacterial lawns were resuspended in M9 minimal medium, spotted onto NGM plates, and allowed to dry overnight prior to transferring worms. *E. coli* ∆*rpoS* and *E. coli* WT were grown aerobically overnight in Lennox Broth (LB) in a 37 °C incubator shaking at 220 RPM, seeded on NGM plates and dried for two days prior to worm plating. PolyQ aggregates were quantified as previously described [12]. In brief, age-synchronized worms (L1 stage) were plated on NGM plates. After four days, worms were washed off plates using M9, washed once if necessary to remove bacteria, and transferred to 96-well plates. The plates were frozen at −20 °C for 1–2 days. Unless otherwise stated, aggregates were quantified using Leica MZ10F Modular Stereo Microscope equipped with CoolLED pE300lite 365 dir mount STEREO with filter set ET GFP-MZ10F. 

### 4.4. ProteoStat Assay

The assay was carried out as previously described [12]. Briefly, *P. aeruginosa*, *P. corporis*, and *E. coli* OP50 were grown in Reinforced Clostridial Broth (RCB) overnight in a 37 °C incubator shaking at 220 RPM. *E. coli* ∆*rpoS* and *E. coli* WT were grown aerobically overnight in LB in a 37 °C incubator shaking at 220 RPM. A 200 μL fraction of each overnight culture adjusted to OD_600_ = 0.5 was transferred into a 96-well plate in sets of nine per each condition or strain. Plates were sealed with parafilm and kept at 26 °C for two days. Following incubation, broth was removed, and wells were washed twice with ddH_2_O. Adhered biofilm was resuspended by vigorous pipetting in 100 μL 1× ProteoStat Assay Buffer. The nine wells per strain/condition were split into sets of three, and cell suspensions from three wells were pooled into a single 300 μL set. Ninety-eight microliters of the pooled fractions were transferred into a black, clear-bottom 96-well plate, and 2 μL of 1x ProteoStat Detection Reagent was added to each well. The content was mixed by pipetting followed by a 15-min incubation at room temperature in the dark. Fluorescence was read at excitation wavelength 550 nm and emission wavelength 600 nm in a Tecan Infinite 200 Pro plate reader. 

### 4.5. Butyrate Treatment

*P. aeruginosa* and *E. coli* cultures were grown aerobically in RCB overnight in a 37 °C incubator shaking at 220 RPM. For *C. elegans* aggregate quantification, overnight cultures were seeded on NGM plates supplemented with 12.5 mM sodium butyrate or controls lacking the compound. Cultures were spread evenly across the plate to ensure uniform coverage of the bacterial lawn. Plates were left to dry for two days prior to plating worms. Worms were plated as L1s for 4 days and aggregates were quantified as described in “Aggregate quantification”. For ProteoStat experiments, bacteria were grown in RCB overnight in a 37 °C incubator shaking at 220 RPM, diluted to OD_600_ = 0.5 with RCB, supplemented with 12.5 mM sodium butyrate, kept stationary at 26 °C and assayed after two days as described in “ProteoStat assay”.

### 4.6. Butyrate as Carbon Source Growth Curve

Overnight LB cultures of *P. aeruginosa* and *E. coli* OP50 were pelleted down, washed 3x with M9 minimal medium to remove all traces of media and resuspended in M9 to a final OD_600_ = 1. Two hundred microliters of 1:100 dilution of bacteria in M9 supplemented with either 0.2% glucose or 0.2% sodium butyrate was added to each well of a 96-well plate in duplicates or triplicates. As *E. coli* OP50 is a uracil autotroph, 4 μg/mL uracil was also supplemented in the M9 minimal medium used for this strain. The plate was incubated at 37 °C in a Tecan Infinite 200 Pro plate reader for 24 h. Every 30 min, the plate was shaken for 5 s, and OD_600_ readings were taken. 

### 4.7. Protein Gel and Coomassie Staining

Insoluble and soluble protein fractions were obtained and a protein gel was visualized as previously described [30] with a few modifications. In brief, *E. coli* ∆*rpoS* and *E. coli* WT were grown for 14–16 h in LB in a 37 °C incubator shaking at 220 RPM. Four milliliters of bacteria at OD_600_ = 1 was pelleted at 3000× *g* for 15 min at 4 °C. Seven hundred fifty microliters of supernatant were removed from each tube and kept on ice for later analysis of secreted proteins. All additional supernatant was then removed and discarded, and 50 μL of ice-cold lysis buffer (10 mM potassium phosphate pH 6.5, 1 mM EDTA, 20% sucrose, 1 mg/mL lysozyme, and 50 u/mL benzonase) was used to resuspend cell pellets. Samples were incubated for 30 min on ice followed by flash-freezing by dipping tubes in 200-proof ethanol at −80 °C for 10 min. After thawing on ice, 360 μL of ice-cold Buffer A (10 mM potassium phosphate pH 6.5, 1 mM EDTA) was added only to the cell samples. Cell samples were transferred to lysing matrix A tubes (lysing tubes had the large bead removed and only glass particles left in the tubes). Four microliters of 1 mM PMSF was added to each sample. Cell samples were homogenized in the Beadbug™3 Microtube Homogenizer for 30 min at 1400 RPM and 4 °C. The samples were then incubated on ice for 5 min to settle glass particles. Two hundred microliters of cell lysate were transferred to new microcentrifuge tubes followed by a 20-min centrifugation at 16,000× *g* at 4 °C. The supernatant was removed and kept separately as a measure for non-secreted soluble proteins. 

The following procedures were applied to both the samples for secreted and non-secreted insoluble proteins: samples were resuspended in 200 μL of Buffer A and centrifuged at 16,000× *g* at 4 °C for 20 min. After removing all supernatant, samples were resuspended in 200 μL of Buffer B (10 mM potassium phosphate pH 6.5, 1 mM EDTA, 2% Nonidet P-40) and centrifuged at 16,000× *g* at 4 °C for 20 min. All supernatant was removed. Samples were resuspended in 200 μL of Buffer A, and centrifuged at 16,000× *g* at 4 °C for 20 min. After removing all supernatant, samples were resuspended in 100 μL total volume of 1x sample buffer and 1x reducing agent and incubated for 5 min at 95 °C. Samples were stored at −20 °C until use. 

The procedure for soluble proteins was the following: 200 μL of the sample was mixed with 50 μL 100% trichloroacetic acid. Samples were incubated for 10 min at 4 °C; a white precipitate formed at this time. Samples were centrifuged at 21,000× *g* for 5 min at 4 °C. After removing all supernatant, samples were washed with 200 μL of ice-cold acetone. Samples were then centrifuged at 21,000× *g* for 5 min at 4 °C. This acetone wash was repeated twice, for a total of three washes. After removing all supernatant after the final wash, microcentrifuge tubes were heated to 37 °C with lids open for no more than 5 min to facilitate evaporation of excess acetone. Samples were resuspended in 100 μL total volume of Criterion XT™ 1x sample buffer and 1x reducing agent, boiled for 5 min at 95 °C, and stored at −80 °C until use.

Both insoluble and soluble fractions, for secreted and non-secreted proteins, were run on a single gel. The Criterion™ Electrophoresis Cell was filled with 1 L of Criterion XT™ MOPS buffer and loaded with a Criterion™ XT Precast Gel. Eighteen microliters of each sample were loaded into each well. The gel was run until the indicator band reached the bottom of the gel. The gel was incubated in Fairbanks A staining solution at room temperature for 30 min on an orbital shaker. The gel was de-stained overnight using Fairbanks D de-staining solution on an orbital shaker. 

The gel was imaged on the iBright FL1000 Imaging System. The resultant image was analyzed using the FIJI image processing plugin. Details are in “Image J Quantification”. 

### 4.8. Image J Quantification

The gel image was loaded into FIJI and cropped down to include only the relevant area. Then, FIJI’s native rolling-ball background subtraction at a radius of 400 pixels was applied to the image. Finally, the mean gray value of each column was measured using FIJI’s Measure tool. Corresponding data points were normalized to the value of the WT.

### 4.9. Gentamicin Treatment

*E. coli* OP50 cultures grown aerobically in LB overnight in 37 °C incubator shaking at 220 RPM were used to seed NGM plates supplemented with 0.2 mg/mL gentamicin or controls lacking the antibiotic. Cultures were spread evenly across the plate to ensure uniform coverage of the bacterial lawn. Plates were left to dry for two days prior to plating worms. The aggregate quantification was carried out as described in “Aggregate quantification” with the exception that aggregates in *C. elegans* exposed to gentamicin and non-gentamicin control were counted using Zeiss Axiovert S100 equipped with Achrostigmat 10X Ph1 phase-contrast infinity microscope objective (0.25NA), Chroma EGFP/FITC long-pass filter set (19002), and a mercury lamp. For ProteoStat experiments, bacteria were grown in RCB overnight in a 37 °C incubator shaking at 220 RPM, diluted to OD_600_ = 0.5 with RCB, supplemented with a final concentration of 1 μg/mL gentamicin, kept stationary at 26 °C and assayed after two days as described in “ProteoStat assay”.

### 4.10. Gentamicin Growth Curve

Serial dilutions of gentamicin (12.5–0 μg/mL) were prepared in a 96-well plate in 50 μL of LB per well. An equal volume of overnight *E. coli* OP50 culture diluted to 1:500 of OD_600_ = 1 was added to each well, diluting the final concentrations of gentamicin by half (6.25–0 μg/mL). The plate was incubated at 37 °C in a Tecan Infinite 200 Pro plate reader for 20 h. Every 30 min, the plate was shaken for 5 s, and OD_600_ readings were taken. Assays were performed in triplicates.

### 4.11. Gentamicin Viability Assay

Bacteria were seeded onto either plain or 0.2 mg/mL gentamicin-supplemented NGM plates. The plates were left to dry for two days at room temperature. The bacterial lawns were washed off with 100 μL of LB, transferred into a sterile tube, and 10 μL was inoculated into LB for an overnight culture. The next day, growth was confirmed in the overnight cultures and photographs were taken.

### 4.12. Congo Red Assay

Curli production was determined by Congo Red plate assay as previously described [12]. In brief, Congo Red agar plates were made using LB medium without salt (5 g yeast extract and 10 g Bacto tryptone per liter of ddH_2_O). After autoclaving, filter-sterilized solutions of Congo Red and Brilliant Blue G250 were added for final concentrations of 50 μg/mL and 1 μg/mL, respectively. The resulting solution was stirred, and plates were poured. Overnight cultures were adjusted to OD600 = 1, and 25 μL of each strain was spotted and spread in a quadrant of a Congo Red agar plate. The plate was incubated at 26 °C for two days, 4 °C for another two days, and photographs were taken.

### 4.13. Quantification and Statistical Analysis

Data were considered statistically significant when *p* < 0.05 was obtained by Student’s *t*-test or one-way analysis of variance (ANOVA) followed by multiple comparison Dunnett’s post-hoc test, as indicated in figure legends. Asterisks denote corresponding statistical significance (* *p* < 0.05, ** *p* < 0.01, *** *p* < 0.001, **** *p* < 0.0001. Error bars represent standard error of the mean (SEM). Statistical analysis was performed using GraphPad Prism 9.3.1 software.

## Figures and Tables

**Figure 1 ijms-23-04807-f001:**
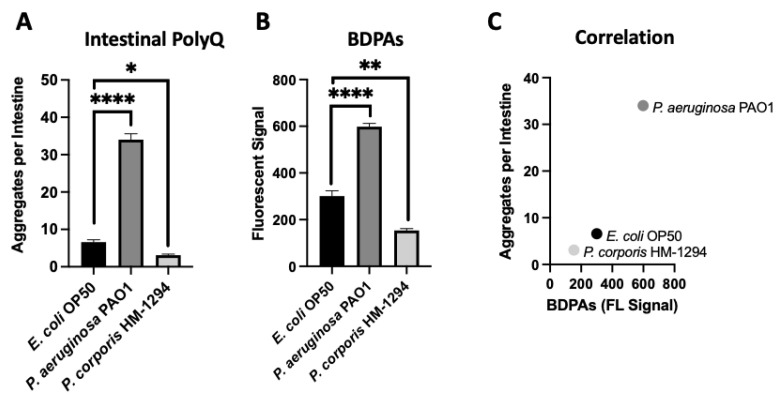
The extent of polyQ aggregation in the *C. elegans* gut upon bacterial colonization correlates with the relative abundance of BDPAs. (**A**) The effect of *P. aeruginosa* PAO1 and *P. corporis* HM-1294 colonization of the *C. elegans* intestine on polyQ aggregation. Data are represented as the average number of polyQ aggregates per *C. elegans* intestine. Each data point is an average of three independent experiments with a total of 60 worms. (**B**) ProteoStat assay quantification of total protein aggregates produced by *E. coli* OP50, *P. aeruginosa* PAO1, and *P. corporis* HM-1294. Data are represented as the average fluorescent signal per bacterial strain stained with ProteoStat. Each data point is an average of two independent experiments with three internal replicates per run. (**C**) A positive correlation between the average number of polyQ aggregates and BDPAs. Error bars represent standard error of the mean (SEM). Statistical significance was calculated using one-way analysis of variance (ANOVA) followed by multiple comparison Dunnett’s post-hoc test (* *p* < 0.05, ** *p* < 0.01, **** *p* < 0.0001).

**Figure 2 ijms-23-04807-f002:**
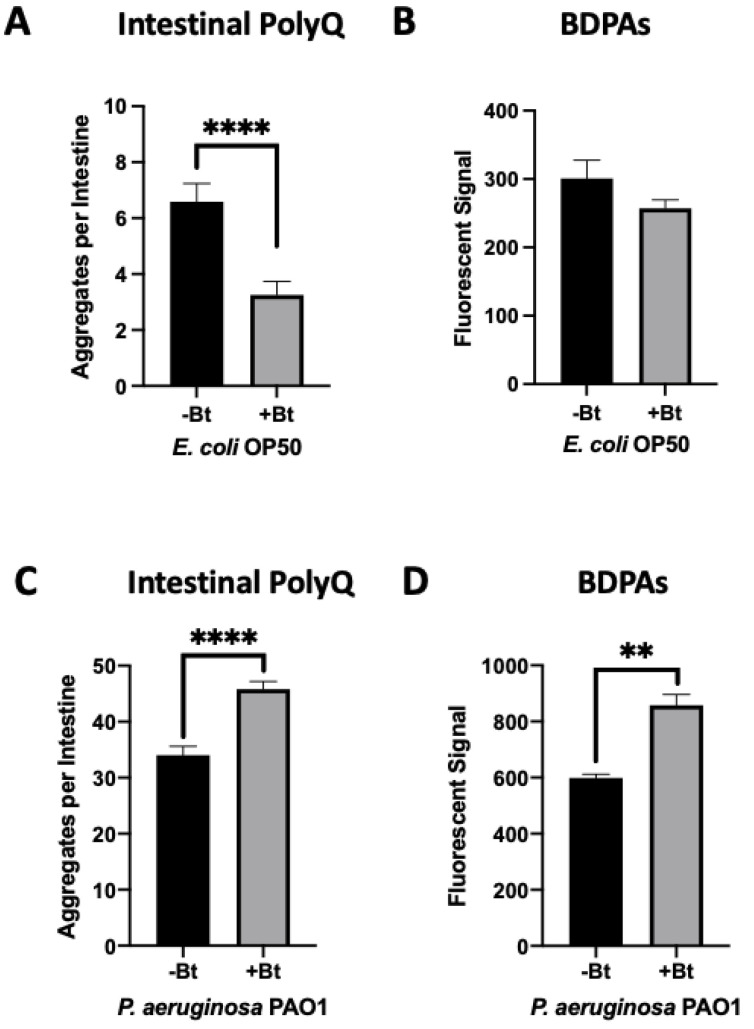
Low-concentration butyrate increases the abundance of *P. aeruginosa* BDPAs, which contributes to polyQ aggregation in *C. elegans*. The effect of (**A**) *E. coli* OP50 and (**C**) *P. aeruginosa* PAO1 colonization of the *C. elegans* intestine on polyQ aggregation in the absence (-Bt) and presence (+Bt) of 12.5 mM butyrate. Data are represented as the average number of polyQ aggregates per *C. elegans* intestine. Each data point is an average of three independent experiments with a total of 60 worms. ProteoStat assay quantification of total protein aggregates produced by (**B**) *E. coli* OP50 and (**D**) *P. aeruginosa* PAO1 in the absence (-Bt) and presence (+Bt) of butyrate. Data are represented as the average ProteoStat fluorescent signal per bacterial strain per treatment. Each data point is an average of two independent experiments with three replicates per run. (**A**–**D**) Error bars represent SEM. Statistical significance was calculated using Student’s *t*-test (** *p* < 0.01, **** *p* < 0.0001).

**Figure 3 ijms-23-04807-f003:**
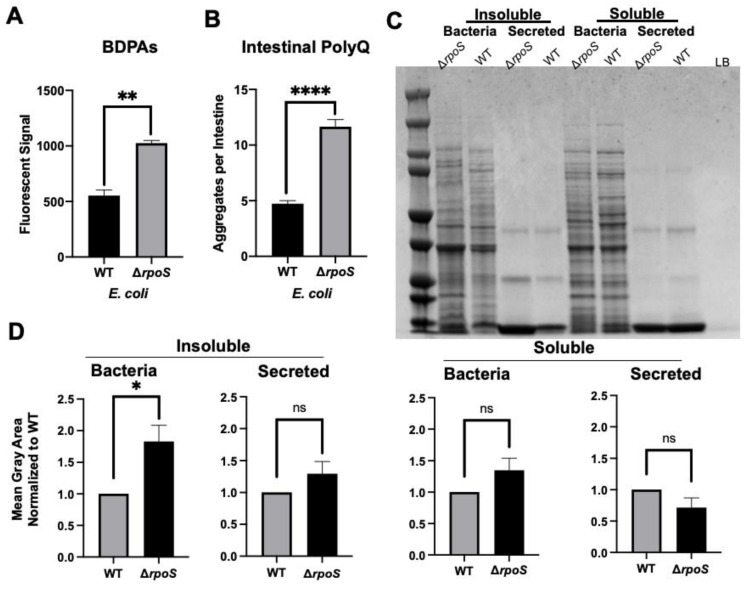
*E. coli* Δ*rpoS* has increased levels of BDPAs and its colonization of the *C. elegans* intestine induces polyQ aggregation. (**A**) ProteoStat assay quantification of total protein aggregates produced by *E. coli* wild-type (WT) and *rpoS* deletion (∆*rpoS*). Data are represented as the average ProteoStat fluorescent signal per bacterial strain. Each data point is an average of two independent experiments with three internal replicates per run. (**B**) The effect of *E. coli* wild-type (WT) and *rpoS* deletion (∆*rpoS*) colonization of the *C. elegans* intestine on polyQ aggregation. Data are represented as the average number of polyQ aggregates per *C. elegans* intestine. (**C**) An SDS-PAGE of insoluble and soluble fractions from whole-cell lysate (Bacteria) and supernatant (Secreted) proteins. Image is representative of the results obtained from three independent experiments. Each data point is an average of three independent experiments with a minimum of 84 worms. The empty lane contains Lennox broth medium only (LB). (**D**) Image-J (FIJI) quantification of proteins from (**C**). Data are representative of three independent experiments. (**A**,**B**,**D**) Error bars represent SEM. Statistical significance was calculated using Student’s *t*-test (* *p* < 0.05, ** *p* < 0.01, **** *p* < 0.0001).

**Figure 4 ijms-23-04807-f004:**
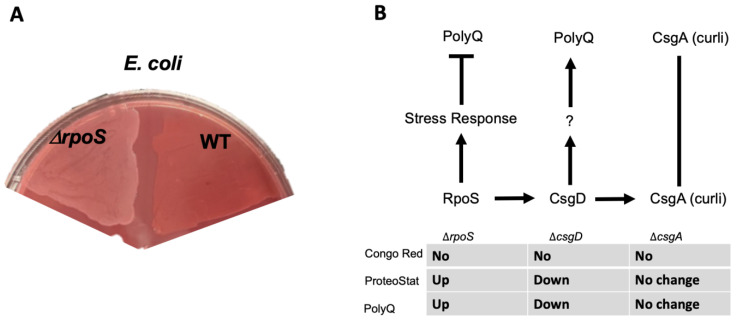
*E. coli* strain lacking RpoS induces polyQ aggregation through a mechanism independent of amyloid production. (**A**) Congo Red staining of amyloids in *E. coli* lacking RpoS (Δ*rpoS*) and wildtype (WT). (**B**) A summary of the results indicating the absence of amyloids as assessed by the absence of Congo Red staining (No), change in the BDPAs assessed by the ProteoStat dye, and the effect on polyQ aggregation upon colonization of the *C. elegans* intestine. We previously showed that unknown product(s) (?) regulated by CsgD contributes to polyQ aggregation and that CsgA (major structural subunit of curli) had no effect on polyQ [12].

**Figure 5 ijms-23-04807-f005:**
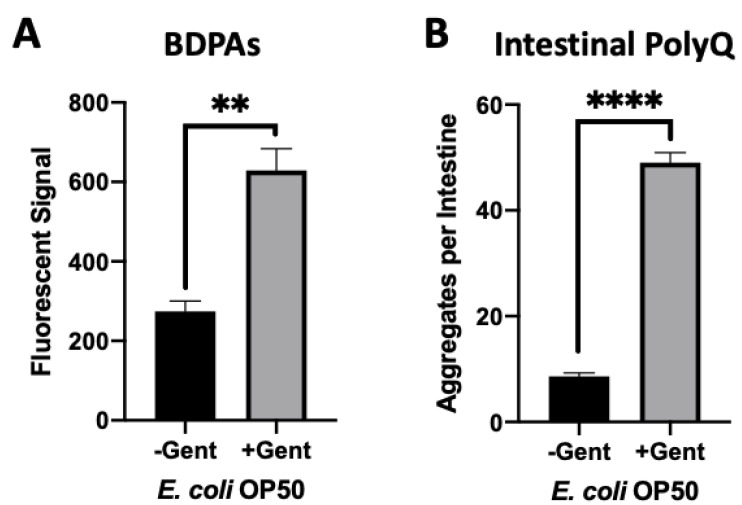
Gentamicin-induced disruption of bacterial proteostasis increases polyQ aggregation in the *C. elegans* intestine. (**A**) ProteoStat assay quantification of total protein aggregates produced by *E. coli* OP50 in liquid culture in the absence (-Gent) or presence (+Gent) of 1 µg/mL gentamicin, a non-effective concentration (Appendix A). Data are represented as the average ProteoStat fluorescent signal per bacterial strain per treatment. Each data point is an average of two independent experiments with three internal replicates per run. (**B**) Intestinal polyQ aggregation in *C. elegans* colonized with *E. coli* OP50 in the absence (-Gent) or presence (+Gent) of 200 µg/mL, a sub-lethal concentration (Appendix A). Data are represented as the average number of polyQ aggregates per *C. elegans* intestine. Each data point is an average of three independent experiments with a minimum of 90 worms. (**A**,**B**) Error bars represent SEM. Statistical significance was calculated using Student’s *t*-test (** *p* < 0.01, **** *p* < 0.0001).

**Figure 6 ijms-23-04807-f006:**
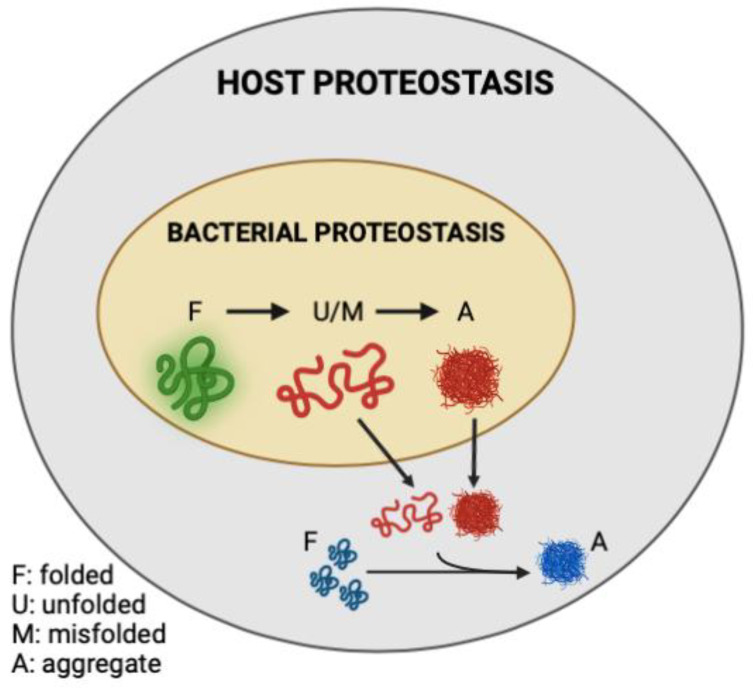
Model figure demonstrating the contribution of misfolded and aggregation-prone bacterial proteins to the disruption of host proteostasis.

## Data Availability

All data supporting the findings of this study are available from the corresponding author on request.

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
