# Peer review of "Bacteria-Derived Protein Aggregates Contribute to the Disruption of Host Proteostasis"

_ijms, 2022, doi:10.3390/ijms23094807_

Round 1

Reviewer 1 Report

The authors assert that bacterial products, which are non-amyloidal in nature (which they refer to as BDPAs), are proteotoxic to the host, in this case C. elegans.   In their experiments, they use three different strains of bacteria, which are producing different levels of BDPAs, which positively correlated with different degrees of polyQ aggregation in the host’s intestine.  They used gentamicin, butyrate and genetic knockout strains to induce changes to the levels of BDPAs and ultimately to polyQ aggregation.

It is curious that, in the introduction, the authors chose to mention the economic cost of neurodegenerative diseases as rationale for scientific research, without even mentioning the suffering of patients and families.

The authors do not convincingly demonstrate that BDPAs cause proteotoxicity. PolyQ aggregation in the intestine is frankly not evidence of proteotoxicity. At a minimum, they should demonstrate phenotypic changes to the host and/or infiltration of these species and PolyQ aggregation into non-intestinal tissues.

The non-amyloidal nature of BDPAs in not confirmed in every experiment. It is possible that the bacterial species involved are amyloid.

Bacterial monocultures are useful in deciphering relevant contributions of bacteria but they are not really clinically relevant as the host’s microbiomes are diverse. Mixed bacterial cultures should have been included in each experiment to observe potential interaction between bacteria and their impact on BDPAs

It is not clear why E.coli is used as the control strain. Is it because it conveniently produces average levels of BDPAs? A control without any bacteria should have been included in every experiment to show that PolyQ aggregation is absent in the host in the absence of bacteria.

The authors only used one concentration of chemical interventions (Butyrate and gentamicin). A dose-response effect is always desirable in scientific experiments.  Tunicamycin could have been used in addition to gentamicin to provide further mechanistic support for this effect.

 They show, or cite, no evidence that butyrate is used as an energy source by P.aeruginosa.

They do not hypothesize on a mechanism of BDPA release by bacteria.

Reviewer 2 Report

The work of A.C. Walker et al. is built up on previous research developed by the Dr. Daniel M. Czyz group (reference 12). The present manuscript highlights the important role of bacteria-derived protein aggregates (BDPAs) on the development of protein conformational diseases (PCDs). This study conducts fluorescence assays (made by ProteoStat kit) to quantify the number of aggregates per worm intestine for several bacteria lines. The gathered findings may be relevant for the examined field. The results achieved are well-discussed during the main body of the reported manuscript. The scientific paper is well written. In my opinion the present manuscript is innovative and the methodological approached used matches with the scope of International Journal of Molecular Sciences. For the above described reasons, I recommend the publication in International Journal of Molecular Sciences once the following minor remarks will be fixed:

--------

RESULTS

Result section is well-structured and clearly explained. Nevertheless, authors should pay attention to the quality of the figures because some of them are blurry images and potential readers could have some difficulties to read them (e.g. Fig. 1 and Fig.2 are blurry. In Fig. 3d, the lettering of “Soluble secreted” is neat in contrast to the rest of the Figure. Fig.4, Fig.5 and Fig.6 are okay).

--------

DISCUSSION

Authors worked with worms as model system. Authors broadly suggest during the entire manuscript body the interconnection between the BDPAs observed from the fungi located inside worm intestines and the potential influence of these BDPAs on the development of neurodegenerative diseases. Worms do not have hematoencephalic barrier as humans. I consider that a small statement should be included in the discussion section regarding this aspect being referred with relevant bibliography: “The present study is focused on the quantification of BDPAs on worm model systems. The most significant outcomes achieved can be extrapolated to humans since it has been deeply demonstrated that protein aggregates can cross the human blood-brain barrier [Adiutori, R.; Puentes, F.; Bremang, M.; Lombardi, V.; Zubiri, I.; Leoni, E.; Aarum, A.; Sheer, D.; McArthur, S.; Pike, I.; Malaspina, A. Analysis of circulating protein aggregates as a route of investigation into neurodegenerative disorders. Brain Commun. 2021, 3 (3), fcab148. https://doi.org/10.1093/braincomms/fcab148]”.

--------

CONCLUSIONS

Conclusions section is optional. However, it would be desirable to add this section in the present manuscript to remark the most significant outcomes of the submitted work.

--------

BIBLIOGRAPHY

The bibliography is not in the proper format of IJNS journal. Authors must take care of this aspect and deeply revise this section.

Reviewer 3 Report

The authors of this manuscript investigated the role of bacteria-derived protein aggregates (BDPA) in host proteostasis. Especially the authors focused on the role of BDPA in host intestinal polyQ aggregates. The authors identified that P. aeruginosa distinctively induced polyQ aggregation and it is more enhanced by low-concentration of butyrate. However, although the authors showed several new findings regarding intestinal polyQ aggregation by P. aeruginosa, mechanism of this phenomenon is poorly suggested.

Main concerns.

  1. In the first part of this study, the authors showed P. aeruginosa-derived BDPAs may induces intestinal polyQ of C. elegans. And the last part, the authors revealed that E. coli delat_rpoS also showed similar results with P. aeruginosa. However, it is not direct evidence occurred in P. aeruginosa. The authors should mentioned the connection between P. aeruginosa and E. coli delta_rpoS.
  2. The authors suggested crude mechanism of RpoS-CsgD-CsgA pathway. However, the authors haven't shown any experimental evidence of this signaling pathway. 
  3. In the previous study by the authors showed that butyrate (50mM) reduced several kinds of bacteria including P. aeruginosa BDPA and polyQ aggregation. But in this study the authors suggested that low-concentration butyrate (12.5mM) enhances only P. aeruginosa BDPAs and polyQ aggregation (but not in other kinds of bacteria including E. coli). This is interesting finding but unusal event. And even the authors did not show any enhanced P. aeruginosa BDPAs in the previous study (10 and 25mM of butyrate). Therefore, it is important to intensively describe the reason of this result. And why 12.5mM and 50mM of butyrate show opposite result in P. aeruginosa? Although the authors previously reported, it is better to show the effect of both low and high dose of butyrate in P. aeruginosa BDPA and polyQ aggregation in this study.

Round 2

Reviewer 1 Report

The manuscript is minimally revised and minimally improved. The authors make the case that additional experiments are not required because they were either performed previously by their group, published by other groups, or beyond the scope of this project. 

Reviewer 3 Report

The authors addressed all my concerns.